# MULTI-TASK LEARNING BY A TOP-DOWN CONTROL NETWORK

## ABSTRACT

As the range of tasks performed by a general vision system expands, executing multiple tasks accurately and efficiently in a single network has become an important and still open problem. Recent computer vision approaches address this problem by branching networks, or by a channel-wise modulation of the network feature-maps with task specific vectors. We present a novel architecture that uses a dedicated top-down control network to modify the activation of all the units in the main recognition network in a manner that depends on the selected task, image content, and spatial location. We show the effectiveness of our scheme by achieving significantly better results than alternative state-of-the-art approaches on four datasets. We further demonstrate our advantages in terms of task selectivity, scaling the number of tasks and interpretability.

Code is supplied in the Supplementary material and will be publicly available.

## 1 INTRODUCTION

The goal of multi-task learning is to improve the learning efficiency and increase the prediction accuracy of multiple tasks learned and performed in a shared network.

In recent years, several types of architectures have been proposed to combine multiple tasks training and evaluation. Most current schemes assume task-specific branches, on top of a shared backbone (Figure 1a) and use a weighted sum of tasks losses for training (Chen et al., 2017; Sener & Koltun, 2018). Having a shared representation is more efficient from the standpoint of memory and sample complexity (Zhao et al., 2018), but the performance of such schemes is highly dependent on the relative losses weights that cannot be easily determined without a "trial and error" search phase (Kendall et al., 2018).

Another type of architecture (Zhao et al., 2018; Strezoski et al., 2019) uses task-specific vectors to modulate the feature-maps along a feed-forward network, in a channel-wise manner (Figure 1b). Channel-wise modulation based architecture has been shown to decrease the destructive interference between conflicting gradients of different tasks (Zhao et al., 2018) and allowed Strezoski et al. (2019) to scale the number of tasks without changing the network. Here, both training and evaluation use the single tasking paradigm: executing one task at a time, rather than getting responses to all the tasks in a single forward pass. Executing one task at a time is also possible by integrating task-specific modules along the network (Maninis et al., 2019). A limitation of using task-specific modules (Maninis et al., 2019) or of using a fixed number of branches (Strezoski et al., 2019), is that it may become difficult to add additional tasks at a later time during the system life-time.

We propose a new type of architecture with no branching, which performs a single task at a time with no task-specific modules. Our model is trained to perform a set of tasks $(\{t_i\}_{i=1}^T)$ one task at a time. The model receives two inputs: the input image, and a learned vector that specifies the selected task $t_k$ to perform. It is constructed from two main parts (Figure 1c): a main recognition network that is common to all tasks, termed below BU2 (BU for bottom-up), and a control network that modifies the feature-maps along BU2 in a manner that will compute a close approximation to the selected task $t_k$. As detailed below, the control network itself is built from two components (Figure 1d): a top-down (TD) network that receives as inputs both a task vector as well as image information from a bottom-up stream termed BU1 (Figure 1d). As a result, the TD stream combines task information with image information, to control the individual units of the feature-maps along BU2. The modification of units

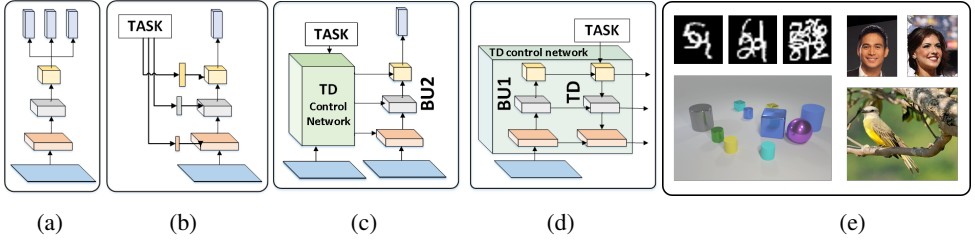

(a)  (b)  (c)  (d)  (e)

Figure 1: **Schemes and Datasets:** (a) Multi-branched architecture, task-specific branches on a top of a shared backbone. (b) Channel-wise modulation architecture, uses task vectors to modulate the feature maps along the main network. (c) **Our architecture uses a top-down (TD) control network** and modifies the featuremaps along the recognition net (BU2) element-wise, according to the image and to the current task. (d) **The internal structure of the control network:** image information, extracted by BU1, is combined with task information and accumulated by the TD stream, to control the units along BU2. (e) Images examples with their corresponding tasks. upper part: M-MNIST, the task is to recognize all the digits, CELEB-A, example tasks are the classification of a smile, sunglasses or earrings. lower part, left: CLEVR, an example task is to recognize the material of the cylinder to the right of the blue cube. right: CUB200, an example task is to identify the color of the bird's neck.

activity in BU2 therefore depends on the task to perform, the spatial location, and the image content extracted by BU1. As shown later, the task control by our approach becomes highly efficient in the sense that the recognition network becomes tuned with high specificity to the selected task $t_k$.

Our contributions are as follow:

a. Our new architecture is the first to modulate a multi-task network as a function of the task, location (spatial-aware) and image content (image-aware). All this is achieved by a top-down stream propagating task, image and location information to lower levels of the bottom-up network.

b. Our scheme provides scalability with the number of tasks (no additional modules / branches per task) and interpretability (Localization of relevant objects at the end of the top-down stream).

c. We show significantly better results than other state-of-the-art methods on four datasets: Multi-MNIST (Sener & Koltun, 2018), CLEVR (Johnson et al., 2017), CELEB-A (Liu et al., 2015) and CUB-200 (Welinder et al., 2010). Advantages are shown in both accuracy and effective learning.

d. We introduce a new measure of task specificity, crucial for multi-tasking, and show the high task-selectivity of our scheme compared with alternatives.

## 2 RELATED WORK

Our work draws ideas from the following research lines:

**Multiple Task Learning (MTL)**   Multi-task learning has been used in machine learning well before the revival of deep networks (Caruana, 1997). The success of deep neural networks in the performance of single tasks (e.g., in classification, detection and segmentation) has revived the interest of the computer vision community in the subject (Kokkinos, 2017; He et al., 2017; Redmon & Farhadi, 2017). Although our primary application area is computer vision, multi-task learning has also many applications in other fields like natural language processing (Hashimoto et al., 2016; Collobert & Weston, 2008) and even across modalities (Bilen & Vedaldi, 2016).

Over the years, several types of architectures have been proposed in computer vision to combine the training and evaluation of multiple tasks. First works used several duplications (as many as the tasks) of the base network, with connections between them to pass useful information between the tasks (Misra et al., 2016; Rusu et al., 2016). These works do not share computations and cannot scale with the number of tasks. More recent architectures, which are in common practice these days, assume task-specific branches on top of a shared backbone, and use a weighted sum of losses to train them. The joint learning of several tasks has proven beneficial in several cases (He et al., 2017), but can also decrease the accuracy of some of the tasks due to limited network capacity, the presence of uncorrelated gradients from the different tasks and different rates of learning (Kirillov et al., 2019). A naive implementation of multi-task learning requires careful calibration of the relative losses of the different tasks. To address these problem several methods have been proposed: 'Grad norm' (Chen

et al., 2017) dynamically tunes gradient magnitudes over time to obtain similar rates of learning for the different tasks. Kendall et al. (2018) uses a joint likelihood formulation to derive task weights based on the intrinsic uncertainty in each task. Sener & Koltun (2018) applies an adaptive weighting of the different tasks, to force a pareto optimal solution on the multi-task problem.

Along an orthogonal line of research, other works suggested to add task-specific modules to be activated or deactivated during training and evaluation, depending on the task at hand. Liu et al. (2019) suggests task specific attention networks in parallel to a shared recognition network. Maninis et al. (2019) suggests adding several types of low-weight task-specific modules (e.g., residual convolutional layers, squeeze and excitation (SE) blocks and batch normalization layers) along the recognition network. Note that the SE block essentially creates a modulation vector, to be channel-wise multiplied with a feature-map. Modulation vectors have been further used in Strezoski et al. (2019) for a recognition application, in Cheung et al. (2019) for continual learning applications and in Zhao et al. (2018) for a retrieval application and proved to decrease the destructive interference between tasks and the effect of catastrophic forgetting.

Our design, in contrast, does not use a multi-branch architecture, nor task-specific modules. Our network is fully-shared between the different tasks. Compared to Zhao et al. (2018), we modulate the feature-maps in the recognition network both channel-wise and spatial-wise, also depending on the specific image at hand.

**Top-Down Modulation Networks**   Neuroscience research provides evidence for a top-down context, feedback and lateral processing in the primate visual pathway (Gazzaley & Nobre, 2012; Gilbert & Sigman, 2007; Lamme et al., 1998; Hopfinger et al., 2000; Piëch et al., 2013; Zanto et al., 2010) where top-down signals modulate the neural activity of neurons in lower-order sensory or motor areas based on the current goals. This may involve enhancement of task-relevant representations or suppression for task-irrelevant representations. This mechanism underlies humans ability to focus attention on task-relevant stimuli and ignore irrelevant distractions (Hopfinger et al., 2000; Piëch et al., 2013; Zanto et al., 2010).

In this work, consistent with this general scheme, we suggest a model that uses top-down modulation in the scope of multi-task learning. Top down modulation networks with feedback, implemented as conv-nets, have been suggested by the computer vision community for some high level tasks (e.g., re-classification (Cao et al., 2015), keypoints detection (Carreira et al., 2016; Newell et al., 2016), crowd counting (Sam & Babu, 2018), curriculum learning (Zamir et al., 2017), etc.) and here we apply them to multi-task learning applications.

## 3  APPROACH

A schematic illustration of our network is shown in Figures 1c and 2a, and explained further below. A control network is used in this scheme to control each of the units in the main recognition network (BU2) given the task and the current image. In practice, this scheme is implemented by using three separate sub-networks (two of them identical) with lateral inter-connections. We next describe the network architecture and implementation in detail.

**Overall structure and information flow**   Our model contains three essentially identical subnetworks (BU1, TD, BU2), with added lateral connections between them. The networks BU1, BU2 share identical weights. The network receives two inputs: an input image (to both BU1, BU2), and a task specification provided at the top of the TD network by a one-hot vector selecting one of k possible tasks. The processing flows sequentially through BU1, TD (in a top-down direction), BU2, and the final output is produced at the top of BU2. In some cases, discussed below, we used an additional output (object locations) at the bottom of TD. During the sequential processing BU1 first creates an initial image representation. The TD creates a learned representation of the selected task that converts the one-hot vector to a new form (task embedding), and propagates it down the layers. On the way down the TD stream also extracts relevant image information from the BU1 representation via the BU1-TD lateral connections. Finally, the TD stream controls the BU2 network, to apply the selected task to the input image via the TD-BU2 lateral connections.

**Bottom-Up streams**   The BU streams use a standard backbone (such as Resnet, VGG, LeNet, etc.), which is usually subdivided into several stages followed by one or more fully-connected layers,

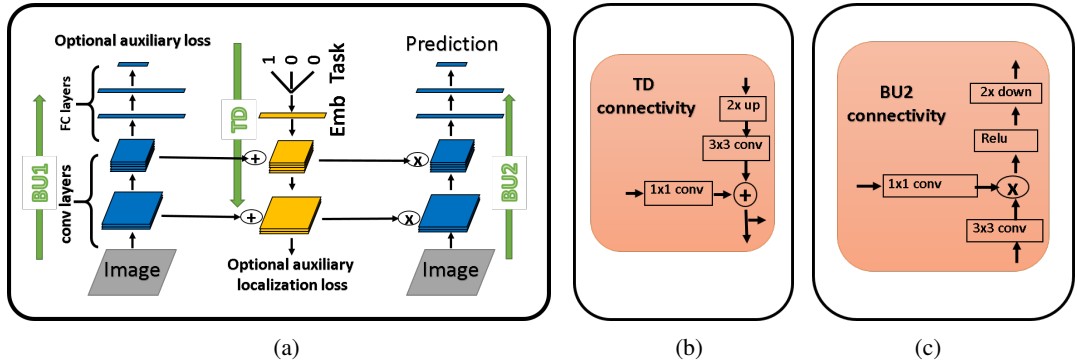

Figure 2: (a) The model architecture. The model consists of three identical networks, BU1, TD, BU2 (e.g., each one a Resent-18). BU1 and TD together constitute the TD-control network in figure 1c. The input consists of the input image (to BU1, BU2), and a task specification at the top of TD. The flow in TD goes from top to bottom; the processing flows sequentially through BU1, TD, BU2, with final output at the top of BU2. (b,c) The lateral connections implemented as 1x1 convolutions. Connections from BU1 to TD (b) are additive, from TD to BU2 (c) are multiplicative.

including the final classifier. The lateral connections between streams are placed at the end of each stage, connecting between tensors of the same sizes, allowing element-wise modifications.

**Top-down stream** The TD stream we use (unless stated otherwise) is a replica of the BU stream in terms of number of layers, type of layers (convolutional / residual) and number of channels in each layer. The downsampling layers (used in the BU stream) are replaced with upsampling layers (nearest neighbour interpolation layers). This design allows us to immediately extend any given BU backbone to our scheme, and it gives good results in our comparison, but the optimal TD structure is subject to future studies. The TD stream has two inputs: the selected task at its top, and inputs from BU1 via lateral connections. The selected task is usually specified by a one-hot vector, which is transformed via learnable weights, into a learned task-representation tensor (called the task embedding, 'Emb' in Figure 2a) that serves as an input to the TD stream.

**lateral connections** For the lateral connections, we experimented extensively with different types, and based on the results we selected two types of connections, one for BU1-to-TD, which is additive, the second for TD-to-BU2, which is multiplicative, shown in Figures 2b and 2c. More details and ablation studies of the lateral connections are given in the Supplementary.

**Auxiliary losses** The use of three sub-networks (BU1, TD, BU2) suggest the natural use of auxiliary losses at the end of the BU1 or TD streams. In the scope of multi-task learning, the TD auxiliary loss can be used to train the extraction of useful spatial information such as the detection of task-relevant objects. This issue is further discussed in Section 4.2 where we demonstrate the use of a localization loss in the last TD feature-map. We show that applying a localization loss allows us to obtain a task-dependent spatial map in inference time, helping interpretability by locating objects of interest.

**Training & evaluation** During training, the learning optimizes all the weights along the BU and TD streams, shared by all tasks, as well as the task specific embedding parameters. Learning uses a standard backpropagation, as the full model forms an end-to-end trainable model.

In training time, the network is supplied with an input image and a selected task, drawn at random from the different tasks. During testing, the different tasks are applied sequentially to each test image.

We used in our implementation shared weights between BU1 and BU2 streams. The main motivation for this design was to allow in future applications a multi-cycle use of the model, by using the BU and TD streams iteratively. With this broader goal in mind, our scheme can also be seen as an unfolded version of a BU-TD recurrent network for one and a half cycles, which is the minimal cycles that allow an image-aware modification process.

## 4 EXPERIMENTS

We validated our approach on four datasets (Multi-MNIST, CLEVR, CELEB-A and CUB-200) with tasks ranging from low-level (e.g., colors) to higher-level (e.g., CLEVR configurations, facial features) recognition, and from simple (e.g., classification by location) to more complex tasks (e.g., classification by combined attributes and spatial relations). We compared our approach to alternatives in terms of prediction accuracy, scaling to a larger number of tasks and task selectivity.

### 4.1 DATASETS & TASKS

**Multi-MNIST**    (Sabour et al., 2017) is a version of the MNIST dataset in which multiple MNIST images are placed on a grid with some spatial overlaps, previously used in Sener & Koltun (2018). We used 2x1, 2x2, 3x3 grids; several training examples are shown in Figure 1e. We used Multi-MNIST to test performance on two sets of tasks: (a) recognizing a digit at a selected location ('by loc', e.g., to recognize the digit in the upper-right location) and (b) recognizing a digit which is right to another digit ('by ref', e.g., to recognize the digit to the right of the digit '7').

**CLEVR**    is a synthetic dataset, consisting of 70K training images and 15K validation images. The dataset includes images of simple 3D objects, with multiple attributes (shape, size, color and material) together with corresponding (question-answer) pairs. We used the CLEVR dataset to test performance on sets of 'by ref' tasks, scaling the number of tasks up to 1645, with a fixed model size. We created multiple tasks by randomly choosing 40, 80, 160 and 1645 queries about an attribute of an object to the (left, right, up, down) of a referred object. An example task is: "What is the color of the object to the left of the metal cylinder?" (metallic cylinder is the referred object).

**CELEB-A**    is a set of real-world celeb face images, intensively used in the scope of MTL (e.g., Sener & Koltun (2018), Strezoski et al. (2019)) on attribute classification tasks. The dataset consists of 200K images with binary annotations on 40 face attributes related to expression, facial parts, etc.

**CUB-200**    is a fine-grained recognition dataset that provides 11,788 bird images of 200 bird species, previously used in (Strezoski et al., 2019). We used CUB-200 to test performance on real-world images with low-level features, and to demonstrate our use of interpretability. An example task is to recognize the color of the bird's crown.

The datasets and corresponding tasks are illustrated in Figure 1e and further discussed in the Supplementary material.

### 4.2 IMPLEMENTATION DETAILS

We performed five randomly initialized training runs for each of our experiments, and present average accuracy and standard deviation.

We used LeNet, VGG-11, VGG-7 and resnet-18, as our BU backbone architectures for the Multi-MNIST, CLEVR, CELEB-A and CUB-200 experiments respectively. We used the Adam optimizer, and performed learning rate search over $\{1e^{-5}, 1e^{-4}, 1e^{-3}, 1e^{-2}\}$ on a small validation set; the main hyperparameters (number of epochs, learning rate and batch size) are shown in Table 1.

Table 1: Main Hyperparameters

| dataset | epochs | l.r. | b.s |
|---------|--------|------|-----|
| M-MNIST | 100 | $1e^{-3}$ | 512 |
| CLEVR | 100 | $1e^{-4}$ | 128 |
| CELEB | 50 | $1e^{-3}$ | 512 |
| CUB | 200 | $1e^{-4}$ | 128 |

We trained the full model end-to-end, using cross entropy loss at the end of BU2. In some of the experiments of CLEVR and CUB-200 datasets we added an auxiliary loss at the end of the TD stream. The target in this case is a 224x224 mask, where a single pixel, blurred by a Gaussian kernel (s.d. 3 pixels) was labeled as the target location. Training one task at a time, we minimized the cross-entropy loss over the 224x224 image at the end of the TD softmax output (which encourages a small detected area), for each visible ground-truth annotated object or part. This auxiliary loss, allows us, at inference, to create task-dependent spatial maps of detected objects; examples of interest are shown in figure 4 and in the Supplementary material. For a fair comparison, we also trained another version of the channel modulation architecture with an additional regression loss, calculated by a FC layer at the top of the network, using the same ground truth annotations.

Database details, full architecture description, more hyperparameters and an analysis of the number of parameters in the architectures can be found in the Supplementary material.

Table 2: Mean of 5 repetitions on M-MNIST. Our architecture consistently achieves highest accuracies.

| ALG | 2 digits #P | Av. Acc | 4 digits #P | Av. Acc | 9 digits #P | by loc Av. Acc | 9 digits #P | by ref Av. Acc |
|---|---|---|---|---|---|---|---|---|
| Single task | x2 | 96.46 | x4 | 94.15 | x9 | 86.62 | x10 | 50.33 |
| Uniform scaling | x1.12 | 95.30 | x1.37 | 90.71 | x1.99 | 74.80 | x2.11 | 29.90 |
| grad-norm | x1.12 | 95.44 | x1.37 | 91.39 | x1.99 | 74.66 | x2.11 | 30.03 |
| kendall | x1.12 | 95.51 | x1.37 | 91.48 | x1.99 | 74.56 | x2.11 | 29.36 |
| mult-obj-opt | x1.12 | 95.81 | x1.37 | 91.24 | x1.99 | 75.13 | x2.11 | 29.82 |
| task-routing | x1.004 | 95.12 | x1.008 | 92.09 | x1.018 | 80.52 | x1.021 | 40.00 |
| ch-mod | x1.004 | 95.87 | x1.008 | 91.38 | x1.018 | 76.56 | x1.021 | 32.69 |
| ch-mod (extended) | x1.35 | 96.30 | x1.36 | 92.96 | x1.37 | 79.81 | x1.38 | 38.57 |
| **ControlNet (ours)** | x1.29 | **96.67** | x1.32 | **94.64** | x1.39 | **88.07** | x1.40 | **72.25** |

## 4.3 COMPARISONS:

We compared our method with existing alternatives from three main approaches (listed in Table 2): (i) A 'Single task' approach, where each task is performed by its own network, (ii) a 'Multi-branched' approach, where the sum of the individual losses is minimized (in 'Uniform scaling' the losses are equally weighted, whereas in 'mult-obj-opt' (Sener & Koltun, 2018), 'kendall' (Kendall et al., 2018) and 'grad-norm' (Chen et al., 2017) the weights are dynamically tuned) and (iii) a 'Modulation' approach, where channel-wise vectors modulate the recognition network in several of the net's stages ('ch-mod' (Zhao et al., 2018) uses learnable weights and 'task-routing' (Strezoski et al., 2019) uses constant binary weights). For a fair comparison with our approach we placed the task embeddings ('modulation' approach) and the lateral connections from the TD ('ControlNet, ours' approach) in the same recognition network locations followed by a single branch for final recognition (the original implementation of 'task-routing' uses multiple branches).

## 4.4 RESULTS

Table 2 summarizes our results on the Multi-MNIST experiment with 2, 4 and 9 digits. We show the average accuracy of all tasks based on 5 experiments for each row. The '$\#P$' column shows the number of parameters as a multiplier of the number of parameter in a standard LeNet architecture. Detailed results with standard deviations and number of parameters are specified in the Supplementary material. Our method achieves significantly better results than the other approaches, even compared with the single-task baseline. Scaling the number of tasks increases the accuracy gap almost without additional parameters. The 'by ref' test (10 tasks) proved to be more difficult (lower accuracies), and shows a significant gap between our approach and other methods. Extending the channel-modulation scheme to a wider backbone ('ch-mod(extended)' in the table, with 15 and 25 channels featuremaps) with roughly the same number of parameters as in our scheme, maintains a large accuracy gap.

Our quantitative results for the CLEVR, CELEB-A and CUB-200 experiments are summarized in Table 3. The '$\#P$' column shows the number of parameters as a multiplier of the number of parameter used in the corresponding BU architecture. Experiments that used localization ground-truth data are indicated with $\sqrt{}$ on the 'loc' column. The results show better accuracy of our scheme compared to all baselines. Table 3a shows that our results on CLEVR, 40 tasks setting, surpass other methods by a significant margin. Our advantage in the CELEB-A and CUB-200 experiments is smaller than in the former tests, possibly due to database biases (e.g., color of different bird's parts are highly correlated) and the diverse appearance of relevant attributes in these databases.

Table 3: Performance on CLEVR, CELEB-A and CUB200. Our approach yields better accuracy than alternatives

(a) CLEVR

| CLEVR | loc | #P | Av. Acc. $\pm$ std |
|---|---|---|---|
| Single task | $\times$ | x40 | $\times$ |
| Uni-sc | $\times$ | x21.03 | 67.66 $\pm$0.81 |
| Uni-sc | $\sqrt{}$ | x21.04 | 73.56 $\pm$0.31 |
| ch-mod | $\times$ | x1.002 | 87.05 $\pm$0.59 |
| ch-mod | $\sqrt{}$ | x1.003 | 89.87 $\pm$0.40 |
| task-routing | $\sqrt{}$ | x1.003 | 69.76 $\pm$0.21 |
| **ControlNet** | $\sqrt{}$ | x1.56 | **96.83** $\pm$0.08 |

(b) CELEB-A

| CELEB-A | #P | Av. Acc. $\pm$ std |
|---|---|---|
| Single task | x40 | $\times$ |
| Uni-sc | x32.64 | 90.36 $\pm$0.03 |
| ch-mod | x1.013 | 90.06 $\pm$0.02 |
| task-routing | x1.013 | 89.80 $\pm$0.04 |
| **ControlNet** | x1.15 | **90.46** $\pm$0.03 |

(c) CUB-200

| CUB200 | loc | Av. Acc. $\pm$ std |
|---|---|---|
| Single task | $\times$ | 74.34 $\pm$0.07 |
| Uni-sc | $\times$ | 77.49 $\pm$0.05 |
| ch-mod | $\times$ | 79.87 $\pm$0.14 |
| ch-mod | $\sqrt{}$ | 79.91 $\pm$0.18 |
| task-routing | $\sqrt{}$ | 80.04 $\pm$0.22 |
| **ControlNet** | $\sqrt{}$ | **80.89** $\pm$0.09 |

## 4.5 EXPERIMENTS DISCUSSION

We discuss below additional aspects of the experiments and general conclusions.

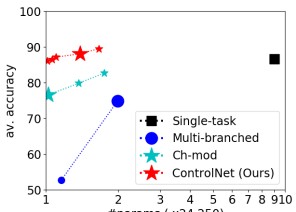

Figure 3: Our architecture achieves higher performance for a similar number of parameters.

**Accuracy vs. model size tradeoff** Figure 3 demonstrates the average accuracy of the 9-class experiment as a function of the number of parameters in four types of architectures; top-left is better. Large markers in the figure correspond to the fixed architectures used in the experiments above. Within each family, parameters can be changed by changing (uniformly) the number of channels along the network. Small markers correspond to modified network sizes: wider LeNet architectures (termed above 'ch-mod(extended)'), or to reduced TD implementations (using TD streams with 1, 4 or 6 channels along its feature-maps). Exact design choices are summarized in the Supplementary material. Our control network architectures correspond to the highest (red) curve in the plot, indicating higher performance for a similar number of parameters. A similar comparison for the CLEVR dataset is reported in the Supplementary material.

**Scaling the number of tasks** Table 2 above shows the accuracies of the Multi-MNIST experiment with the 2, 4 and 9 tasks datasets. Note that as the number of digits increases, the task also increases in difficulty due to the digits overlap. Increasing the number of tasks increases the accuracy gap compared with alternative models with a similar or even larger number of parameters.

Table 6a below shows the accuracies of gradually increasing the number of tasks up to 1645 on the CLEVR dataset. Compared with alternatives, our results are scalable with the number of tasks, with a smaller decrease in performance for 1645 tasks. The uniform scaling approach cannot scale above 80 tasks due to memory restrictions, and did not take part in this experiment. Overall, the results show that our scheme deals better in performing multiple tasks in limited capacity, the performance gap increases with the number of tasks and complexity and the scheme benefits from the contribution of spatial information and image content.

**Adding Tasks** A general question of interest in mutli-task learning is whether the tasks need to be pre-defined, compared with the possibility of adding at least some tasks at a later stage to an already trained model. We used in this task an extension of the M-mnist 'by-ref' experiment. Specifically, our architecture has been trained and evaluated on the defined 'by ref' tasks, excluding the task involving the digit '9' (9 tasks in all). We then extended the embedding layer and trained it, while keeping the rest of the model fixed, on the new task examples. Table 4 shows the results. The obtained accuracy for the added digit '9' task are 64.68%, other tasks mean accuracy remains unchanged (74.89%). The new task accuracy is lower than the mean, but shows significant learning, compared with the pre-trained accuracy of 16.3%. The accuracy of all other tasks is unaffected (avoiding 'catastrophic forgetting') without requiring any further training of the previous tasks.

Table 4: Adding task (Acc%)

|  | before | after |
|---|---|---|
| existing tasks | 74.89 | 74.89 |
| added task | 16.30 | 64.68 |

**Tasks heterogeneity** Our model performs multiple tasks in the same network by applying activation-modifications according to the task and to the image at hand. A question of interest is whether the set of tasks is limited to a homogenous set, such as similar classification tasks (as in the M-MNIST experiment), or extends to more heterogeneous tasks. In particular, performing both recognition and pixel-labeling tasks in the same network using task selection schemes (e.g., channel modulation, task-routing, ours) has not been studied in the past.

Table 5: **Hetrogenous tasks:** executing recognition and segmentation with / without tasks selection.

|  | loc inst. | cls/seg inst. | # branches | 9 digits - by loc CLS (Acc) | SEG (IOU) | # branches | 9 digits - by ref CLS (Acc) | SEG (IOU) |
|---|---|---|---|---|---|---|---|---|
| Multi-branched | × | × | 18 | 74.10 | 59.07 | 20 | 31.68 | 36.29 |
| ControlNet | ✓ | × | 2 | 88.40 | 65.34 | 2 | 73.83 | 46.73 |
| ch-mod | ✓ | ✓ | 2 | 76.67 | 61.16 | 2 | 46.96 | 38.97 |
| ControlNet | ✓ | ✓ | 2 | 88.53 | 67.46 | 2 | 75.53 | 48.07 |

We extended our proposed architecture to perform both recognition tasks (producing class labels) and segmentation tasks (producing a spatial map), guided by the TD instruction provided to the network. The instruction is composed of two parts: selecting a digit, either by location or by reference (the 'by loc' and the 'by ref' vector in the M-MNIST experiments above), and then applying either recognition or segmentation, using a two slots one-hot-vector (results in Table 5 row 4). We compared this task selection with three alternatives. The first includes no instruction, implemented with an individual branch for each task (top row, multi-branched architecture). The second performs both segmentation and recognition together, implemented with two simultaneously executed branches (second row in the table). The final compares with channel modulation (row 3). To deal with both recognition and segmentation tasks we used two separate output branches (one producing one of 10 class labels, the other producing a 28*28 map) rather than a single one. The branches were used according to the task instruction: using one at a time when classification/segmentation instruction is given (rows 3, 4), or using both branches simultaneously when no such instruction is provided (row 2). Results show that our model with the selected branch (row 4) performs better than all alternatives. A direct comparison with the alternative of using the two branches together, as in standard (un-instructed) branching models (second row), reveals that the selected branch increases in performance, and at the same time, the results of the un-selected branch significantly decreases in performance, by 10-30%.

**Ablations and the use of image content** We compared our scheme, which uses image information (via BU1) and performs full tensor element-wise modification of the feature maps (of BU2) to the channel modulation scheme (which performs channel modulation with no image content information) with roughly the same number of parameters, shown in Table 6b, top row (ch-mod (extended)). We also tested two ablations of our network: one without the BU1 stream, removing the image-content contribution (TD, second row) and second without the TD stream, where the task is supplied to both BU streams, concatenated to the image data (BU, third row, details in Supplementary). We conducted the experiments on two sets of tasks applied to the 9-location MNIST: classification by location ('by loc') and classification by reference ('by ref'), and on two sets of tasks used in the CLEVR dataset (40 and 1645 tasks), which are inherently a 'by ref' tasks, using reference objects. Table 6b below shows that ControlNet achieves significantly better results than its ablations, with a gap that increases as the complexity or the number of tasks increase. The comparison between our model (4th row) and our ablated model (2nd row), which does not use the image content in the modification process, shows the significant advantage of using image content information. The addition of BU1 (which shares weights with BU2) improves the accuracy substantially with almost no additional parameters.

**Task Selectivity** Task-selectivity is likely to be a crucial property of architectures that execute one task at a time (such as ch-mod, ours) in order to fully utilize the network resources for the selected task. We defined a task-selectivity measure by comparing the prediction accuracy of the model for the selected task vs. non-selected tasks. To make this comparison, we trained readout heads to predict from the final representation produced by the BU2 stream not just the selected task, but all possible tasks. For example, in the 4-digits MNIST, we trained four readout branches on the top of the final representation (trained to produce the selected location) to predict the digit identities at all four locations. We define task-selectivity by the ratio between the accuracy for the selected task (above chance level) and the average accuracy of the non-selected tasks (above chance level). Detailed results are shown in Figure 4a for the multi-MNIST, 4 tasks setting (top) and 9 tasks-setting (bottom).The results show over 90% accuracy for the selected task branch (along the diagonal), and close to chance-level accuracies for all other branches. Figure 4b summarizes the results of the same experiment for the channel modulation architecture, which shows less selectivity. The corresponding selectivity indices for the 4-class case are 26.5 and 8.25 and for the 9-class case 37.23 and 7.97 for our model and channel-modulation respectively. The higher selectivity index of our method is likely

Table 6: (a) **Performance on CLEVR**, Our approach is scalable with the number of tasks with an increasing gap over ch-mod. (b) **Test of ablations and image content**. 'im': image contribution.

(a) Scaling the number of tasks

| | number of tasks | | | |
|---|---|---|---|---|
| | 40 | 80 | 160 | 1645 |
| uni-sc | 68.10 | 70.42 | n.a. | n.a. |
| ch-mod | 89.97 | 87.39 | 81.84 | 60.38 |
| ControlNet | **96.85** | **95.60** | **95.57** | **88.83** |

(b) Test of ablations and image content

| method | im | m-mnist by loc | m-mnist by ref | clevr (40) by ref | clevr(1645) by ref |
|---|---|---|---|---|---|
| ch-mod (extended) | × | 79.81 | 38.57 | 91.46 | 73.95 |
| TD | × | 82.22 | 66.86 | 90.10 | 68.12 |
| BU | √ | 82.14 | 55.77 | 93.21 | 73.77 |
| ControlNet | √ | **88.07** | **72.25** | **96.85** | **88.83** |

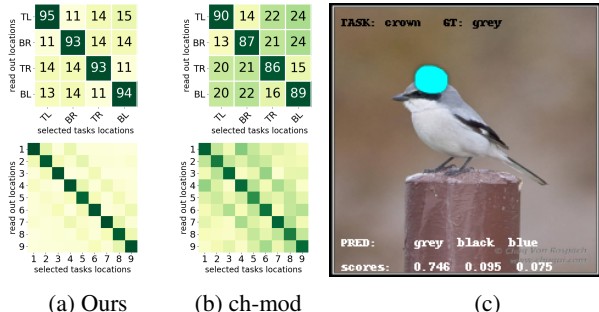 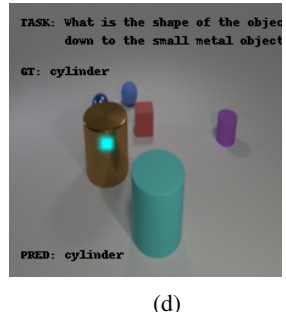

| (a) Ours | (b) ch-mod | (c) | (d) |

Figure 4: (a,b) **Task selectivity.** The columns in each plot are the selected task 4/9 locations, the rows are the readout locations. With high selectivity, off-diagonal recognition will be at chance level. Our scheme (a, selectivity indices 26.5, 37.2) shows higher selectivity than the channel-modulation scheme (b, selectivity indices 8.3, 8.0). (c,d) **Attention maps**, as obtained at the end of the TD stream, highlighted with turquoise; intermediate objects of interest are well localized: (c) CUB-200: What is the color of the bird's crown? (d) CLEVR: What is the shape of the object below the small metal object? Best viewed in color while zoomed-in.

to be related to the increased gap in performance with respect to alternative models in Tables 2 and 3. ==Results in the section on tasks heterogeneity further show that even for heterogeneous tasks, task selection increases the performance of the selected task at the expense of the non-selected tasks.==

**Task-dependent spatial maps**   Using a task dependent localization loss at the end of the TD stream in train time allows us to obtain task-dependent spatial maps in inference time, helping interpretability of the result by locating intermediate objects of interest. Figures 4c and 4d demonstrates the location maps produced by our architecture at inference time. In both examples, the predicted mask/object is well localized (on the crown of the bird or on the object below the small metal object) and the attribute (color or shape) is correctly predicted. In case of wrong result, it is possible to examine whether the error in the attribute was associated with mis-detecting the relevant part. Additional examples of interest and failure cases are shown in the Supplementary material.

## 5 SUMMARY

We described an architecture for multi-task learning, which is qualitatively different from previous models, primarily by the use of the BU1 stream and a TD convolutional stream, which controls the final BU2 stream as a function of the selected task, location and image content. We tested our network on four different datasets, showing improvements in accuracy compared with other schemes, along with scaling the number of tasks with minimal effect on performance, and helping interpretability by pointing to relevant image locations. Comparisons show higher task-selectivity of our scheme, which may explain at least in part its improved performance.

More generally, multiple-task learning algorithms are likely to become increasingly relevant, since general vision systems need to deal with a broad range of tasks, and executing them efficiently in a single network is still an open problem. Our task-dependent TD control network is a promising direction in this field in terms of accuracy and scalability. ==In future work we plan to adapt our architecture to a wider range of applications (e.g., scene understanding, images generation), to a wider range of architectures with higher capacity and to examine the possible combination of branching strategy with our TD task selection approach, extending our heterogeneous tasks example. Our architecture is also potentially useful for problems such as online learning, domain adaptation and catastrophic forgetting, demonstrated in part by our example of adding a new task to an already trained model, and we plan to further explore these central problems in the future.==

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
