# OpenReview forum: "Multi-Task Learning by a Top-Down Control Network"
_ICLR.cc/2021/Conference — Reject_

### Official Review · AnonReviewer1 · 2020-10-25
**BU-TD Network for Multi-task Learning**

**Rating:** 5
**Confidence:** 5

**Review:**

This paper presents a new architecture for multi-task learning that uses a top-down control network to modulate the activations of the main (bottom-up) recognition network. The model is applied to four datasets/tasks: multi-MNIST, CLEVR, CELEB-A, and CUB-200, demonstrating good performance compared with baselines. I have the following comments and questions:

- The proposed architecture requires the equivalent of three forward/backward passes: BU1, TD, and BU2. The number of parameters is used gauge complexity in Table 2 and Figure 3 but FLOPs might be a better metric here since BU1 and BU2 share parameters. How do the models compare in terms of FLOPs to baselines?
- The datasets/tasks used herein are homogenous and therefore straightforward for multi-task learning. How does the proposed architecture fair in a more challenging setting involving heterogeneous tasks, e.g., Misra et al., 2016?
- Re "In some of the experiments of CLEVR and CUB-200 datasets we added an auxiliary loss at the end of the TD stream. The target in this case is a 224x224 mask, where a single pixel, blurred by a Gaussian kernel (s.d. 3 pixels) was labeled as the target location.", How was this mask obtained? How do the models perform without the use of this auxiliary loss? Why was this loss only used in CLEVR and CUB-200?

Minor:
- Add references to first column in Table 2.

To conclude, the proposed architecture is novel, the paper is clear, but the experimental work leaves some questions unanswered.

---

> ### Author Response · Authors · 2020-11-22
> **Response to reviewer #3**
>
> We would like to thank for the thoughtful review. Please find our response to your concerns in the following.
>
> 1. Regarding the amount of computation, the closest model to ours in terms of computations is the extended channel modulation, which uses about the same overall number of parameters and number of operations (flops). This model performs better than the original channel modulation, but performs significantly less well than our model (Tables 2, 4b in the original paper). Other architectures: Single task architecture uses more computations when the number of tasks exceeds 3. The number of operations in Multi-branched architectures linearly increases with the number of tasks. We will finally note that though the number of flops is of large interest, the number of parameters can become, given memory restrictions, of central importance.
>
> 2. Task heterogeneity: This is a very interesting question; aspects of this question are addressed in the general response. In the revised paper we discuss this question both in the description of the added experiment, as well as in the final discussion.
>
> 3. The single pixel coordinates were obtained from the GT annotations of the relevant datasets (CLEVR and CUB-200), and then blurred with a Gaussian kernel and used as an auxiliary loss. This was not used in Celeb-A since the dataset does not provide location annotations. The main reason we used this auxiliary loss is to demonstrate our ability to produce task-dependent spatial maps in inference time, helping interpretability of the result by locating intermediate objects of interest. For a fair comparison we add the exact ground truth information to other baselines (indicated with v on the ‘loc’ column in table 3a and 3c).

---

### Official Review · AnonReviewer4 · 2020-10-28
**More analysis would be appreciated.**

**Rating:** 5
**Confidence:** 3

**Review:**

The paper propose a way to combine image information and task information as controllers for multi-task learning. In this way, the authors expect to extract more task/image specific features in a shared backbone.

The proposed method seems novel and intuitive. Though there are some typos and grammar mistakes, overall the paper is easy to follow.

My major concerns are the followings:
1. the claim that the proposed scheme provides scalability with the number of tasks is stretching: the number of tasks still need to predefine and the number of task heads are not optimized compared to previous algorithms.
2. While there is improvement in performance, but it is not clear what factors causes the improvement. As compared to previous schemes the proposed model needs much more computationally (two bottom up runs and a top-down run) and utilizes more information.
3. All the tasks are classification tasks /discriminative models. It is not demonstrated if this would be working with a mixture of generative/discriminative tasks.

If possible, as the TD and BU's are sharing the same structures, it would be interesting to explore what are learned by visualizing the weights in each channels and layers. and further explore which feature map is heavily used for which specific task. These weights/activation distributions will help us to better understand what is actually learned in the proposed scheme.  The task spatial maps seems to be a good start, but it would be better if the author provide more analysis on intermediate layer activations

---

> ### Author Response · Authors · 2020-11-22
> **Multi-Task Learning by a Top-Down Control Network - response to reviewer #2**
>
> We would like to thank for the thoughtful and constructive review. Please find our response to your concerns in the following:
>
> 1.	Regarding scalability: in the general response we describe an additional experiment where a task is added to the network by learning a new task embedding without changing the main backbone. This illustrates that at least in some cases it is possible to add tasks to the already trained network. There is also evidence for scalability in the results that in our model it is possible to increase the number of tasks (e.g. 1645 in the CLEVR task) and the accuracy remains high compared with alternative approaches. Regarding optimizing the number of heads and number of tasks is an interesting issue. In the general response we show an example in this direction, where we use 2 branches for 20 tasks, for two major families of tasks (discriminative or generative). We finally note that in most previous models the number of branches is simply equal to the number of tasks, with no optimization.
>
> 2.	Where the improvement is coming from:  We analyzed some aspects of this question in our ablation studies that evaluate the contribution of image content and TD context to the results accuracy. As discussed later below (the last point) it will be of interest to study more in the future the specific contributions of different components of the scheme. Regarding the amount of computation, the closest model to ours in terms of computations is the extended channel modulation, which uses about the same overall number of parameters and number of operations (flops). This model performs better than the original channel modulation, but performs significantly less well than our model (Tables 2, 4b in the original paper). In terms of information used we use image and task information, similar to other models. We also use in several cases auxiliary losses, but in the relevant tests, we added these losses to the alternative models, so we believe the comparisons were unbiased.
>
> 3. Combining discriminative and generative tasks. This combination is of general interest and it has not been studied in the multi-task literature in the past. We added an experiment that tests explicitly an aspect of this capacity in our model, described in the general response (first experiment).
>
> 4. Studying and visualizing the roles of different model components:  one possible approach that can be used to address some of these issues is by readout experiments from different components of the system. We applied this in our study of task selectivity in the last layer before the fully connected layers in the network. The higher selectivity index of our method is likely to be related to the increased gap in performance with respect to alternative models. A particularly interesting component to test further would be the representations developed in the task-embedding component of the TD net.

---

### Official Review · AnonReviewer3 · 2020-10-30
**An interesing paper on introducing the top-down information as the supervision for multi-task neural network learning.**

**Rating:** 7
**Confidence:** 3

**Review:**

In this paper a novel top-down control network is introduced for multi-task learning. Different from the traditional bottom-up attention models, the authors introduce a top-down module to modify the activation of recognition network based on different tasks. Specifically，the proposed module consists of three identical networks, which are BU1, TD, BU2 streams. Given the input, the BU1 is firstly trained, and then the TD streams is trained by assigning the specific labels. After that, the BU2 is updated with the top-down parameters. Experimental results demonstrate the effectiveness of proposed model.

Strength:
1. It is interesting to introduce the semantic information from the top layer to guide the feature representation learning.

Weakness:
 Although the experimental results show the better performance on the image classification, there are exist several unclear parts:

1. The definition of multi-task in this paper refers to the different dataset’s classification? Or referring to the different tasks, e.g. localization, classification, and attributes predication. In my opinion, the authors should provide more details on designing the validation experiments. And the proposed model should be tested on different tasks instead of only on the task of visual recognition.
2. Some bottom-up based model e.g. FiLM should be used as the baselines to validate the advantage of introducing the top-down stream.
3. If the top-down stream would make the recognition be sensitive for the visual variations? Or the classification results may be dependent on the number of training samples?

4. If the proposed model would be helpful for transfer learning?

---

> ### Author Response · Authors · 2020-11-22
> **Multi-Task Learning by a Top-Down Control Network - response to reviewer #1**
>
> We would like to thank the reviewer for the positive feedback. We list below answers to specific concerns raised in the review:
>
> 1.    Similar concerns were raised by other reviewers. Please see our answer to all reviewers.
> 2.    Regarding FiLM: There are two differences between FiLM and our approach. First, it keeps an initial backbone unaltered. Second, it uses both multiplicative and additive learned parameters in the modulation part. We tested these two aspects of FiLM adapted to the backbones used in the experiments. (The Resnet-101 network used in the original FiLM is larger than the networks used in our model).
> | | | FiLM | Ours |
> |--|--|--|--|
> |MNIST |2 digits (by loc)| 95.42| 96.67|
> | | 4 digits |92.99 |94.64|
> | |9 digits | 75.75 | 88.07 |
> | |9 digits (by ref) |33.27 | 72.25 |
> |CLEVR | | 62.03 | 96.83 |
>
> 3.    Our recognition results naturally increased with the number of training examples. We found evidence in two results that our model can be trained with a smaller number of examples compared with alternatives. First, our “stress test” results, increasing CLEVR tasks to 1645, in table 4a of the original paper, obtains better results than alternatives by a large gap. In this test, the number of examples is reduced to less than 45 training examples for each task in an epoch.  Second, we compared results of the multi-mnist task with a controlled number of examples for the number ‘3’ in the top-left location (new experiment). The reported accuracy below is for testing specifically instances of images with ‘3’ at the top-left. Learning starts faster and remains higher up to a large number of examples.
>
> |Number of examples|Ours  | chmod |
> |------------------------------|--------|------------|
> | 0 | 0 | 0 |
> | 10 | 0 | 0 |
> |100 | 50.59 | 0 |
> | 1000 | 85.54 | 26.44 |
> | 10000 | 94.85 | 72.87 |
>
> 4.	Use in transfer learning. This is a general interesting issue for future studies which we have not studied in detail. One attractive possibility is to use the learned task embedding in the network, for at least some aspects of transfer leaning. An example along this line is described in experiment 2 of the general response.

---

### Author Response · Authors · 2020-11-22
**common response (all reviewers)**

We thank the reviewers for their thoughtful feedback. We added four follow-up experiments to address concerns expressed in the reviews. In this common response, we describe results of two experiments that addressed general concerns; additional results are described in individual responses.

1.	Tasks heterogeneity

Our models performs multiple tasks in the same network by applying modifications to the activations according to the task and to the image at hand. A recurring question was whether the set of tasks is limited to homogenous tasks, such as similar classification tasks, or extends to more heterogeneous tasks (localization, classification, attributes), or even a mixture of generative/discriminative tasks.
In an added experiment, we extended our proposed architecture to perform both recognition tasks (producing class labels) and segmentation tasks (producing a spatial map), guided by the TD instruction provided to the net. A particular task is selected by a TD instruction using a classification/segmentation vector (a two slots one-hot-vector), concatenated to the existing one-hot-vector for the 9 locations as used in the ‘by loc’ and ‘by ref’ M-MNIST experiments. We implemented in this case two output branches (one producing one of 10 class labels, the other producing a 28*28 map) rather than a single one, but the use of the branches was different from a standard branching model as described further below.
We compared our model with relevant competing alternatives; results for the `by ref’ experiment are shown in the table below. The first comparison is to a 20-branch model, one for each task (recognize the digit right to the digit ‘k’ for 10 possible digits, segmentation/classification); the second is a 2-branch channel modulation; the 3rd is our model with the 2 branches and a digit instruction only (performing both segmentation and classification simultaneously), the 4th is our model with a single task at a time: a selected digit and a classification/segmentation instruction. Our model performs better than all alternatives. In using our model, we compared using the two branches together, as in standard  (un-instructed) branching models (3rd row), with selecting a specific task that uses one of the branches (4th row). By instructing the network, the selected branch (cls/seg) increases in performance, and the un-selected one decreases in performance (by 10-30%). These results also extend our findings on task specificity showing that even for heterogeneous tasks, task selection increases the performance of the selected task at the expense of the non-selected tasks.

|	|								Cls (acc%) |	seg(IOU) |
|--|--|--|
|Multi-branched (20 branches, no digit nor cls/seg  instructions) | 	31.68	|	36.29 |
|Ch-mod (2 branches, digit and cls/seg instructions)		|	46.96	|	38.97 |
|Ours (2 branches, digit instruction only)			|		73.83	|	46.73 |
|Ours (2 branches, digit and cls/seg instructions)	|		75.53	|	48.07 |

The revised paper now includes a description of these results as well as additional ‘by location’ results. It also includes a discussion of task homogeneity, and an addition in the final discussion on the possible use of more than a single branch in the framework of a task-selecting control network.


2.	scalability and adding tasks

A general question of interest in mutli-task learning is whether the tasks need to be pre-defined, compared with the possibility of adding at least some tasks at a later stage. In an added experiment, we demonstrate the ability of the model to add a task to the already trained model. The general idea is to keep the model fixed, and adding a task by training the embedded representation of the added task.
We used in this task an extension of the M-mnist ‘by-ref’ experiment. Specifically, our architecture has been trained and evaluated on the defined ‘by ref’ tasks, excluding the task involving the digit ‘9’ (9 tasks in all). We then extended the embedding layer and trained it only on the new task examples. The obtained accuracy for the added digit ‘9’ task are 64.7%, other tasks mean accuracy remains unchanged (74.89%). The new task accuracy is lower than the mean, but shows significant learning, compared with the pre-trained accuracy of 16.3%. The accuracy of all other tasks is unaffected (avoiding ‘catastrophic forgetting’) without requiring any further training of the previous tasks.

---

### Author Response · Authors · 2020-11-23
**Updated version + summary of changes**

We would like to thank the reviewers for their positive feedback and thoughtful suggestions. We have updated the article, changes are highlighted in yellow.
In summary, compared to our initial submission, we have:
* New paragraph  - adding tasks to an already trained network (Adding task, page 7)
* New paragraph – extending the set of tasks to include both recognition and pixel labeling tasks (Tasks heterogeneity, page 7)
* Addition in the task selectivity paragraph referring to tasks heterogeneity (page 9)
* Addition in the final discussion on the use of more than a single branch in the framework of a task-selecting control network (page 9)

---

### Decision · Program_Chairs · 2021-01-07
**Final Decision**

**Decision:**

Reject

**Comment:**

The paper is very interesting and novel, and all reviewers are of the same opinion.
The main concern, however, is on the experimental section that is limited to image classification benchmarks and that some critical comparisons are missing (e.g. clarify factors that play key role in improvement, more computation and therefore more free parameters, how about non discriminative tasks, etc).
The heterogeneity question is in my opinion only partially answered by the authors but I also feel proper handling of this matter would require a proper multi-task setup and different target for the work.
I also personally find applicability of the approach quite limited, I encourage the authors to further improve their work as I feel that with a proper revision would make a nice contribution for the community.